# Localization of Viral Epitope-Specific CD8 T Cells during Cytomegalovirus Latency in the Lungs and Recruitment to Lung Parenchyma by Airway Challenge Infection

**DOI:** 10.3390/life11090918

**Published:** 2021-09-04

**Authors:** Franziska Blaum, Dominika Lukas, Matthias J. Reddehase, Niels A. W. Lemmermann

**Affiliations:** 1Institute for Virology, Research Center for Immunotherapy (FZI), University Medical Center of the Johannes Gutenberg-University Mainz, 55131 Mainz, Germany; frablaum@uni-mainz.de (F.B.); lemmermann@uni-mainz.de (N.A.W.L.); 2Department of Dermatology, University of Cologne, University Hospital Cologne and Faculty of Medicine, 50937 Cologne, Germany; dominika.lukas@uk-koeln.de

**Keywords:** antigen presentation, CD8 T cells, cytomegalovirus (CMV), effector-memory T cells (TEM), hematopoietic cell transplantation (HCT), interstitial pneumonia, latent infection, lungs, lung parenchyma, memory inflation (MI)

## Abstract

Interstitial pneumonia is a life-threatening clinical manifestation of cytomegalovirus infection in recipients of hematopoietic cell transplantation (HCT). The mouse model of experimental HCT and infection with murine cytomegalovirus revealed that reconstitution of virus-specific CD8^+^ T cells is critical for resolving productive lung infection. CD8^+^ T-cell infiltrates persisted in the lungs after the establishment of latent infection. A subset defined by the phenotype KLRG1^+^CD62L^−^ expanded over time, a phenomenon known as memory inflation (MI). Here we studied the localization of these inflationary T effector-memory cells (iTEM) by comparing their frequencies in the intravascular and transmigration compartments, the IVC and TMC, respectively, with their frequency in the extravascular compartment (EVC), the alveolar epithelium. Frequencies of viral epitope-specific iTEM were comparable in the IVC and TMC but were reduced in the EVC, corresponding to an increase in KLRG1^−^CD62L^−^ conventional T effector-memory cells (cTEM) and a decrease in functional IFNγ^+^CD8^+^ T cells. As maintained expression of KLRG1 requires stimulation by antigen, we conclude that iTEM lose KLRG1 and convert to cTEM after transmigration into the EVC because pneumocytes are not latently infected and, therefore, do not express antigens. Accordingly, antigen re-expression upon airway challenge infection recruited virus-specific CD8^+^ T cells to TMC and EVC.

## 1. Introduction

Human cytomegalovirus (hCMV) is a prototype member of the β-subfamily of the herpes virus family [1]. Whereas primary infection passes mostly undiagnosed without overt clinical symptoms or organ disease when held in check by an intact innate and adaptive immune system in the otherwise healthy, immunocompetent host, unrestricted cytopathogenic tissue infection can lead to multiple-organ failure with often lethal outcome in an immunocompromised host [2,3,4].

A significant public health and economic impact [5,6], and the main argument for the development of a CMV vaccine [7,8,9,10], results from birth defects caused by hCMV in an immunologically immature embryo/fetus after congenital infection following diaplacental transmission of the virus [11,12] after primary or recurrent infection during pregnancy [13].

After clearance of productive CMV infection, a latent infection, briefly referred to as “latency”, is established, which is defined by the presence of replication-competent viral genomes that are silenced at loci critical for completion of the viral replicative cycle so that infectious progeny is not produced [14,15]. However, there is increasing evidence for viral gene expression during latency that does not follow the canonical temporal cascade of gene expression during productive infection [16] and that can impact the cellular secretome to create a latency-modulated microenvironment [17,18,19,20].

Patients at risk of CMV disease following productive reactivation of latent virus are recipients of solid-organ transplantation (SOT), who become immunosuppressed to prevent graft rejection, as well as recipients of hematopoietic cell transplantation (HCT) transiently immunocompromised by hematoablative therapy of aggressive hematopoietic malignancies that are refractory to standard antitumoral therapies. On top of this, in the case of allogeneic HCT with family or unrelated donors differing in major (HLA) and/or minor histocompatibility antigens (mHAg), additional immunosuppressive therapy is applied to prevent graft-versus-host (GvH) disease (for clinical overviews, see [21,22]).

In HCT with either donor or recipient or both being latently infected, hCMV can reactivate from latently infected donor cells of the myeloid hematopoietic lineage [23,24,25,26,27,28], as well as from latently infected cells present in the transplant recipient, presumably including a tissue cell type that is refractory to hematoablative antitumor therapy. Endothelial cells (EC) have been discussed as candidates [29]. Despite routine follow-up of HCT recipients by quantitative PCR to detect virus reactivation at the earliest possible occasion to initiate pre-emptive therapy with antiviral drugs, HCT-associated CMV disease remains a clinical problem due to virus variants that have developed drug resistance [30]. In such patients, immunotherapy by adoptive transfer of virus-specific CD8^+^ T cells has become the last resort to prevent CMV disease [31,32,33,34]. CMV infection of the lungs is the focus of our own work because interstitial CMV pneumonia represents the clinically most feared manifestation of hCMV infection in HCT patients, with an often lethal outcome if treatment fails [35,36,37].

The mouse model of experimental syngeneic, as well as allogeneic, HCT and infection with murine cytomegalovirus (mCMV) has contributed much to the understanding of fundamental common principles of CMV pathogenesis, immune evasion, and immune control, including the intervention by CD8^+^ T-cell-based immunotherapy (reviewed in [38,39,40]). Specifically, as first shown in the model of syngeneic HCT, efficient and timely reconstitution of virus-specific CD8^+^ T cells is crucial for preventing lethal viral pneumonia after acute primary infection [41]. Lung-infiltrating protective CD8^+^ T cells persist in the lungs after clearance of productive infection and the establishment of a latent infection [41]. The observation of an expansion of CD8^+^ T effector-memory cells (TEM) specific for certain viral epitopes in these persisting lung infiltrates [42] initiated extensive research on this phenomenon that is now known as memory inflation (MI) (for reviews, see [43,44,45,46,47,48]). The expanding cells represent a subset of CD8^+^ T cells that is characterized by the cell surface marker phenotype KLRG1^+^CD62L^−^ [49], for which we have proposed the acronym iTEM for “inflationary T effector-memory cells” to distinguish them from the conventional KLRG1^−^CD62L^−^ T effector-memory cells, the cTEM [50]. The iTEM differ from effector cells of the acute immune response by a prolonged life span due to IL15-mediated expression of the antiapoptotic protein Bcl-2 [51]. Importantly, MI represented by an expansion of the iTEM pool depends on a high load of latent viral genomes, which is achieved by systemic infection but not usually by local infection, after which viral replication and spread are limited by immune control in the draining regional lymph node ([50] and references therein).

Since its first description, the maintained expression of KLRG1, and thus the iTEM phenotype, is known to depend on continuous or at least frequent restimulation by antigen [52]. We have recently identified transient and stochastic expression of epitope-encoding viral genes during latency, and thus a sporadic presentation of viral antigenic peptides, as the viral molecular driver of MI [53,54]. This finding explained the independently described stochastic nature of expansion and contraction of viral epitope-specific CD8^+^ T-cell clones during MI [55] and is in perfect accordance with a mathematical model of MI curve fitting, proposing frequent stimulation by antigen to smoothen episodes of pool expansions and contractions [56].

An open question concerns the precise microanatomical site(s) where latently infected cells present antigens for the restimulation of iTEM. Competing though, in our view, not mutually exclusive hypotheses proposed antigen presentation during viral latency to occur in lymphoid tissue [57] or at latently infected cells of the vasculature, for instance at endothelial cells (EC) in the capillaries of the vascular bed of the lungs [58]. In fact, we have previously shown that latent mCMV genomes in the lungs localize to CD31^+^CD146^+^ capillary wall EC [53].

Here we studied the localization of viral epitope-specific iTEM, cTEM, and T central memory cells (TCM) in latently infected lungs. We distinguished three compartments, namely (1) the intravascular compartment (IVC), (2) the transmigration compartment (TMC), which includes CD8^+^ T cells attached to capillary wall endothelium, cells in the process of extravasation, and cells that have arrived in the interstitium, as well as (3) the extravascular compartment (EVC) (Figure 1).

Our data provide reasonable evidence to propose that iTEM receive their restimulation by latently infected cells outside of the lung parenchyma, and convert to cTEM by downregulation of KLRG1 when they localize to the lung parenchyma, which is not latently infected and, hence, does not present viral antigenic peptides for restimulation and maintained expression of KLRG1.

## 2. Materials and Methods

### 2.1. Viruses and Mice

Bacterial artificial chromosome (BAC)-cloned virus MW97.01, derived from BAC plasmid pSM3fr [59,60], is herein referred to as WT.BAC. For challenge infection, recombinant viruses mCMV-IE1-L176A+m164-I265A (referred to as ΔIDE) and mCMV-IE1-A176L+m164-A265I (referred to as ΔIDE_rev) [61] were used. Female BALBc/J mice (8 weeks old, haplotype *H-2^d^*) were purchased from Harlan Laboratories and were housed under specified pathogen-free (SPF) conditions in the Translational Animal Research Center (TARC) of the University Medical Center of the Johannes Gutenberg-University Mainz.

### 2.2. Establishment of Latent Infection after Experimental HCT

Syngeneic hematopoietic cell transplantation (HCT) with 9-week-old female BALBc/J mice as bone marrow cell (BMC) donors and recipients was performed as described ([62] and references therein). In brief, hematoablative conditioning was performed by sublethal total-body γ-irradiation with a single dose of 6.5 Gy. Donor BMC (5 × 10^6^/mouse) were infused into the tail vein of the recipients at 6 h after irradiation, followed by intraplantar infection with 10^5^ plaque-forming units (PFU) of WT.BAC injected into the left hind footpad. Latent infection was routinely confirmed by the presence of viral genomes in tissues in the absence of infectious virus [53].

### 2.3. Airway Challenge Infection of Latently Infected Mice

Mice latently infected with WT.BAC were superinfected with 1 × 10^6^ PFU of viruses ΔIDE or ΔIDE_rev by intratracheal virus application as described [63]. Four days later, localization of virus-specific CD8^+^ T cells to different lung compartments was quantified by cytofluorometric analysis.

### 2.4. Preparation of Compartment-Specific Lung Cells

Latently infected mice were lethally anesthetized by carbon dioxide inhalation. As the first step, bronchoalveolar lavage (BAL) leucocytes were isolated by flushing the airways with DPBS + 2% FCS. [64]. Leucocytes from the bloodstream were isolated from blood taken by heart puncture directly after the BAL. Finally, leucocytes from lung tissue were isolated essentially as described ([50] and references therein). In brief, after perfusion of the lungs via the right ventricle to remove cells from the capillary bed of the lungs, the lungs were excised. Tracheae, bronchi, and pulmonary lymph nodes were discarded, and the lung lobes were minced. Lung tissue from 4 to 5 lungs was digested for 1 h at 37 °C with constant stirring in 15 mL DMEM containing collagenase A (1.6 mg/mL; Roche, Mannheim, Germany) and DNase I (50 µg/mL, Sigma-Merck, Darmstadt, Germany). Mononuclear cells were enriched by density-gradient centrifugation for 30 min at 760× *g* on lymphocyte separation medium Histopaque-1077 (Sigma-Merck).

### 2.5. Cytofluorometric Analyses

Single-cell suspensions were prepared from different lung compartments as described above. Unspecific staining was blocked with unconjugated anti-FcγRII/III antibody (anti-CD16/CD32; clone 2.4G2, BD Bioscience, Heidelberg, Germany). Living cells were detected using Fixable Viability Dye eFluor 780 (ThermoFisher Scientific, Langenselbold, Germany). Cells were specifically stained with the following antibodies for multi-color cytofluorometric analyses: BV421-conjugated anti-CD8α (clone 53-6.7; BioLegend, San Diego, USA), FITC-conjugated anti-KLRG1 (clone 2F1; BioLegend), BV510-conjugated anti-CD45 (clone 30-F11; BioLegend), and PerCP-Cy5.5-conjugated anti-CD62L (clone MEL-14; Thermo Fisher Scientific). Phenotypic characterization of peptide-specific CD8 T cells was performed using PE-conjugated dextramers H-2Ld/YPHFMPTNL (IE1), and H-2Dd/AGPPRYSRI (m164) (Immudex, Copenhagen, Denmark). All cytofluorometric analyses were performed with flow cytometer BD FACSCanto and BD FACSDiva analysis software. For detailed analyses and documentation, FlowJo (version 10.6, BD Biosciences) was used.

### 2.6. Quantitation of Functional Epitope-Specific CD8^+^ T Cells

At the indicated time during viral latency established after HCT and infection, cells isolated from the different lung compartments served as responder cells in an IFNγ-based enzyme-linked immunospot (ELISpot) assay ([65] and references therein). Briefly, to detect functional, epitope-specific CD8^+^ T cells, synthetic peptides were exogenously loaded at a saturating concentration of 10^−7^M on P815 (*H-2^d^*) mastocytoma cells for serving as stimulator cells in the assay. Graded numbers of leucocytes were seeded with the peptide-loaded stimulator cells in triplicate microcultures. After 18 h of coculture, spots, each representing an IFNγ-secreting cell, were counted automatically, based on standardized criteria using ImmunoSpot S4 Pro Analyzer (Cellular Technology Limited, Cleveland, USA).

### 2.7. Antigenic Peptides

Antigenic peptides reported to be presented by MHC class-I molecules K^d^, D^d^, and L^d^ are derived from the mCMV open reading frames (ORF), M105, m123/IE1, m145, and m164 (listed with their amino acid sequences in [62]). Custom peptide synthesis with a purity of >80% was performed by JPT Peptide Technologies (Berlin, Germany).

### 2.8. Statistics

Frequencies (most probable numbers (MPN)) of cells responding in the ELISpot assay, and the corresponding 95% confidence intervals, were calculated by intercept-free linear regression analysis from the linear portions of regression lines based on spot counts from triplicate assay cultures for each of the graded cell numbers seeded [65]. Differences between multiple groups were evaluated using one-way ANOVA with Bonferroni’s post-hoc test and were considered as being significant at significance levels of *p* < 0.05 (*), *p* < 0.01 (**) or *p* < 0.001 (***). All calculations were performed using Graph Pad Prism 6.04, (Graph Pad Software, San Diego, CA, USA).

## 3. Results and Discussion

### 3.1. Reduced Frequency of Viral Epitope-Specific Functional IFNγ^+^CD8^+^ T Cells in the EVC of Latently Infected Lungs

To isolate leucocytes, including CD8^+^ T cells, from the compartments (defined in Figure 1) of latently infected lungs, a gentle bronchoalveolar lavage was performed as a first step to retrieve cells that are only loosely attached to parenchymal epithelial cells of the alveoli, defining the EVC. Blood leucocytes were isolated to represent the IVC, and, as the last step, perfusion-resistant leucocytes were retrieved by enzymatic digestion of lung tissue. This yields intravascular leucocytes that are more firmly attached to EC lining the lung capillaries, cells in the process of transmigration, and extravascular cells localizing to the connective tissue of the interstitium. As these three localizations cannot easily be separated experimentally, they are here collectively referred to as TMC.

Cells with the functional capacity to produce IFNγ upon stimulation with presented viral antigenic peptides were quantitated in an ELISpot assay and were normalized to the proportion of CD45^+^CD8^+^ T cells determined in parallel by cytofluorometric analysis of the respective leucocyte suspensions (Figure 2).

The experiment was originally undertaken with the expectation that viral epitope-specific CD8^+^ T cells floating in the lung capillaries would selectively bind to latently infected EC of the capillary wall and thus would be found enriched in the TMC. At a glance, this was not the case for any of the four viral epitopes tested, although the genes coding for the established immunodominant and MI-driving antigenic peptides IE1 and m164 [62], and also the gene coding for M105, are stochastically expressed in lung EC during latency [53]. Rather, within the 95% confidence intervals, frequencies were comparable in IVC and TMC. In contrast, frequencies were significantly reduced in the EVC for all four viral epitopes tested. Although this shows that functional IFNγ^+^CD8^+^ T cells can, in principle, localize to the lung epithelium also during latent infection of the lungs, their frequency is lower at this site than it is in IVC and TMC.

### 3.2. Localization of Viral Epitope-Specific CD8^+^T-Cell Activation Subsets in Latently Infected Lungs

MI during viral latency is based on an expansion primarily of CD8^+^ iTEM that are characterized by the cell surface marker phenotype KLRG1^+^CD62L^−^. For localizing iTEM in latently infected lungs, we therefore determined the frequencies of iTEM in the three compartments in comparison to the frequencies of KLRG1^−^CD62L^−^ cTEM and KLRG1^-^CD62L^+^ TCM (cytofluorometric data: Figure 3 and Appendix A, summary of results: Figure 4). It should be noted that here we did not further subdivide iTEM by expression of CD127 (IL7-R) because KLRG1^+^CD127^+^ double-positive effector cells (DPEC) do not contribute to MI in latently infected lungs [53].

Regarding absolute cell numbers, the yield of CD8^+^ T cells is generally low in the EVC, which is what one would expect for a nonlymphoid site during a nonacute infection. In particular, in the IVC, we noted an unexpected additional subpopulation of CD45^+^CD8^+^ T cells with the cell surface marker phenotype KLRG1^+^CD62L^+^ (Figure 3). Interestingly, this population was present also among viral epitope-specific CD45^+^CD8^+^IE-TCR^+^ cells. At the moment, we can only speculate that these cells may represent iTEM in a state of transition to memory cells, a state during which KLRG1 is not yet downregulated but CD62L already reacquired. We did here not pursue this subpopulation and its potential function but found its existence worth noting for future work by ourselves or by other investigators.

A compilation of all our cytofluorometric analyses of viral epitope-specific CD8^+^ T-cell subset localization in latently infected lungs (Figure 4) revealed a picture for the iTEM that strikingly parallels the lung compartment distribution shown in Figure 2 for viral epitope-specific functional IFNγ^+^CD8^+^T cells, namely, a deprivation in the EVC compared to IVC and TMC. In contrast, cTEM were enriched in the EVC compared to IVC and TMC. This mirror image in the distribution suggests that iTEM lose KLRG1 and convert to cTEM after they have reached the lung epithelium. This interpretation is supported by the known fact that maintenance of KLRG1 requires continuous or at least frequent restimulation by antigen [52], which is not the case in the lung parenchyma because pneumocytes are not latently infected and thus do not present antigen during latency.

A puzzling question is why we did not find an enrichment of viral epitope-specific cells in the TMC, although latently infected EC in the capillary walls sporadically express epitope-encoding viral genes [53]. One possibility is that we observe a steady-state level in which influx and efflux into and out of this compartment are in balance. Our data cannot decide between the competing hypotheses of MI taking place at the latently infected lung endothelium [58] or in lymphoid tissues, specifically in lung-draining lymph nodes [57]. Cells recognizing antigens presented by latently infected EC of the endothelium need to detach for cell division. So, epitope-specific CD8^+^ T cells stimulated at the endothelium of the pulmonary capillary bed might detach and migrate to the lung-draining lymph nodes for proliferation before they return as iTEM to the lungs for immune surveillance [66].

Finally, as one would have expected for an extralymphoid site, the frequency of viral epitope-specific TCM was low in all three compartments, and TCM were actually undetectable in the EVC.

### 3.3. Acute Airway Challenge Infection Recruits CD8^+^ T Cells into the TMC and EVC in an Epitope-Specific Manner

Data so far have shown that functional IFNγ^+^CD8^+^ T cells (Figure 2) and iTEM (Figure 4) are both deprived in the EVC in latently infected mice. We have explained this by conversion of KLRG1^+^ iTEM into KLRG1^−^ cTEM due to a lack of antigen presentation by lung epithelial cells, which are not cellular sites of latent mCMV infection. If this explanation holds true, it must be postulated that acute infection of lung parenchyma in latently infected mice leads to antigen presentation that restimulates cTEM in the EVC and converts them into iTEM, and infection likely also recruits cells from the IVC into the TMC and further into the EVC. As an approach, we used airway superinfection of latently infected mice, a model that has a clinical correlate since humans latently infected with hCMV can be exposed to infectious virions via the airways upon intimate contact with acutely infected children who shed virus produced in salivary gland epithelial cells into the saliva. To be sure that the virus reaches the alveolar epithelium, we chose intratracheal infection, which also activates migratory CD11b^+^ as well as CD103^+^ dendritic cells [63]. Acute superinfection with a virus mutant in which MI-driving, immunodominant epitopes (IDE) IE1 and m164 are functionally deleted by X9A point mutations of the respective C-terminal amino acid residues (mCMV-ΔIDE) cannot restimulate CD8^+^ T cells specific for these epitopes in mice latently infected with wild-type virus (mCMV-WT.BAC) encoding IE1 and m164. In contrast, restimulation of cells specific for IE1 and m164 should occur after acute superinfection with mCMV-ΔIDE_rev, in which mutations X9A are back-mutated to A9X to restore antigenicity and immunogenicity [61] (Figure 5).

Compared to the compartment distribution of viral epitope-specific CD8^+^ T cells in latently infected lungs prior to an acute airway superinfection (Figure 4), which is characterized by similar frequencies in IVC and TMC and a deprivation in the EVC, the pattern is shifted to a deprivation in the IVC and similar frequencies in TMC and EVC. Notably, the recruitment to TMC and EVC has an epitope-unspecific component seen for IE1-specific and for m164-specific CD8^+^ T cells after airway superinfection with mCMV-ΔIDE not expressing these two epitopes. A likely explanation is a chemokine-mediated recruitment of CD8^+^ T cells to the site of infection, as we have shown previously for mast cell-derived chemokine CCL5 in a model of acute mCMV infection of the lungs [67]. Recruited CD8^+^ T cells in this acute infection model localized histologically to TMC and EVC and controlled the infection of the alveolar epithelium. Added to this chemokine-mediated recruitment is an epitope-dependent recruitment and clonal expansion, as it is revealed by further increased frequencies of IE1-specific and m164-specific CD8^+^ T cells in TMC and EVC upon airway superinfection with mCMV-ΔIDE_rev expressing these two epitopes (Figure 5).

## 4. Conclusions

Our data reveal a caveat about the definition of “tissue resident” CD8^+^ T cells. In particular, in the case of the lungs, in which a widely ramified capillary bed serves for gas exchange, it is important to distinguish between cells floating in the capillaries (IVC), cells adhering to the capillary endothelium, or being in the process of transmigration (TMC), and cells that localize to the lung parenchyma, the alveolar epithelium (EVC). We show here that KLRG1^+^CD62L^−^ iTEM, which account for the phenomenon of “memory inflation” (MI) during latent infection of the lungs, primarily localize to the IVC and TMC, where they receive stimulation by latently infected EC that present antigenic peptides during sporadic episodes of viral gene expression. Upon localization to the alveolar epithelium, which is not latently infected, iTEM become deprived of stimulation by antigen, lose the expression of KLRG1, and convert to KLRG1^−^CD62L^−^ cTEM. In line with this scenario, provision of antigens in the alveolar epithelium by acute airway superinfection recruits epitope-specific CD8^+^ T cells from the IVC to the TMC and also to the EVC.

## Figures and Tables

**Figure 1 life-11-00918-f001:**
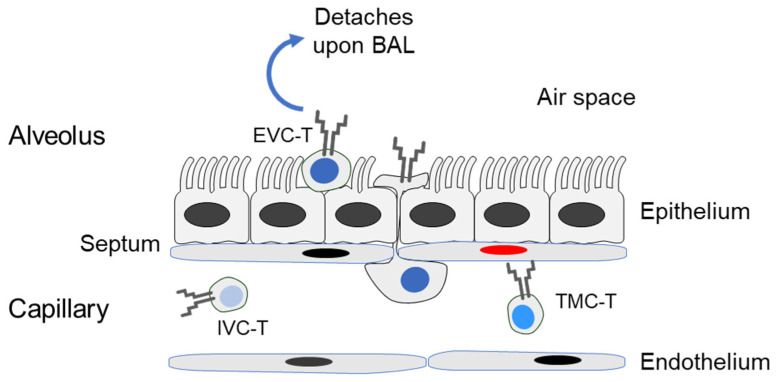
Scheme of the lung capillary-parenchyma interface. (IVC-T) T cell in the intravascular compartment, the capillary blood. (TMC-T) T cell in the transmigration compartment, comprising cells attached to the endothelium, cells in the process of transmigration, and cells in the interstitium. (EVC-T) T cell in the extravascular compartment loosely attached to the alveolar epithelium. (BAL) bronchoalveolar lavage. The red-stained nucleus symbolizes a latently infected endothelial cell.

**Figure 2 life-11-00918-f002:**
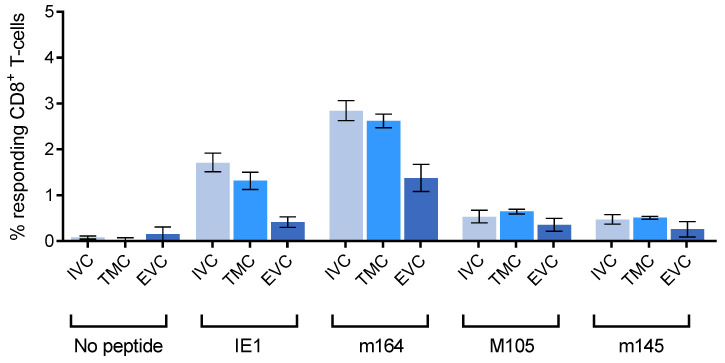
Diminished frequency of epitope-specific functional IFNγ^+^CD8^+^ T cells in the EVC. The analysis was performed 8 months after HCT and infection with mCMV. Bars represent frequencies of CD8^+^ T cells specific for the viral antigenic peptides indicated and localizing to the lung compartments indicated (defined in Figure 1). Error bars represent the 95% confidence intervals (CI) determined by intercept-free linear regression analysis of the ELISpot data. Differences are significant when 95% CI do not overlap.

**Figure 3 life-11-00918-f003:**
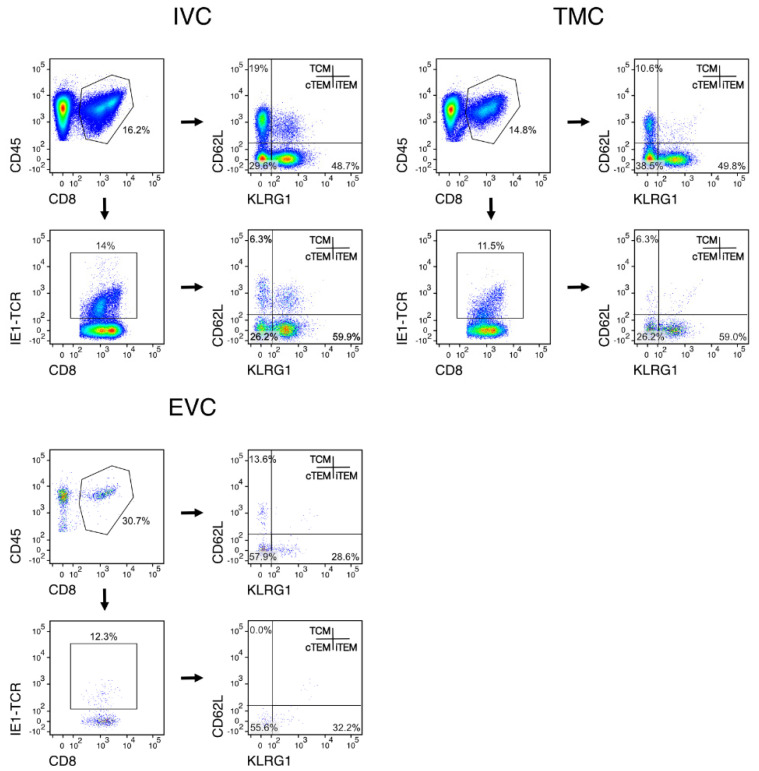
Cytofluorometric relative quantitation of total and IE1 epitope-specific subsets of CD8^+^ T cells in lung compartments. Data correspond to the functional data of the experiment shown in Figure 2. Shown are representative examples illustrating the gating strategy and the definition of activation subsets iTEM, cTEM, and TCM by combination of the cell surface markers KLRG1 and CD62L among pregated total CD45^+^CD8^+^ T cells or among IE1 epitope-specific CD45^+^CD8^+^ T cells that were defined by expression of T-cell receptors specific for IE1 peptide presented by the MHC class-I molecule L^d^. The complete gating strategy, including a gate set on CD45^+^ cells to distinguish between hematopoietic lineage-derived leucocytes and tissue cells, is shown in Appendix A for a representative example of the TMC, in which CD45^−^ tissue cells released by the enzymatic tissue digest could be technically interfering contaminants.

**Figure 4 life-11-00918-f004:**
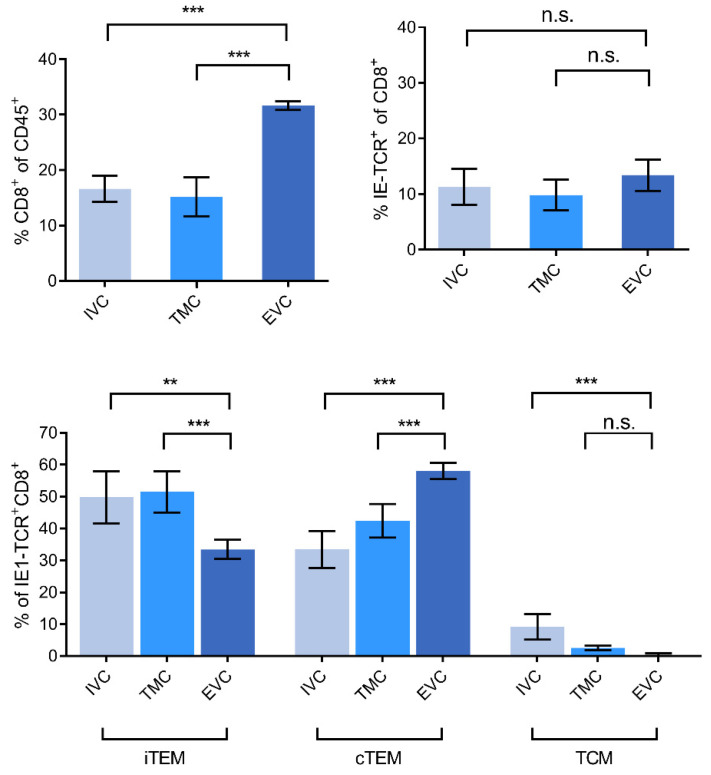
Summary of cytofluorometric localization data. Bars represent mean values, and error bars represent standard deviations for 12 pools of the leucocyte yield of 3 mice per pool (*n* = 12) in the case of IVC and TMC, and for 4 pools of the leucocyte yield from 9 mice per pool (*n* = 4) in the case of the EVC. Significance levels are shown for groups of most interest. (**) p < 0.01. (***) *p* < 0.001. (n.s.) not significant, *p* > 0.05.

**Figure 5 life-11-00918-f005:**
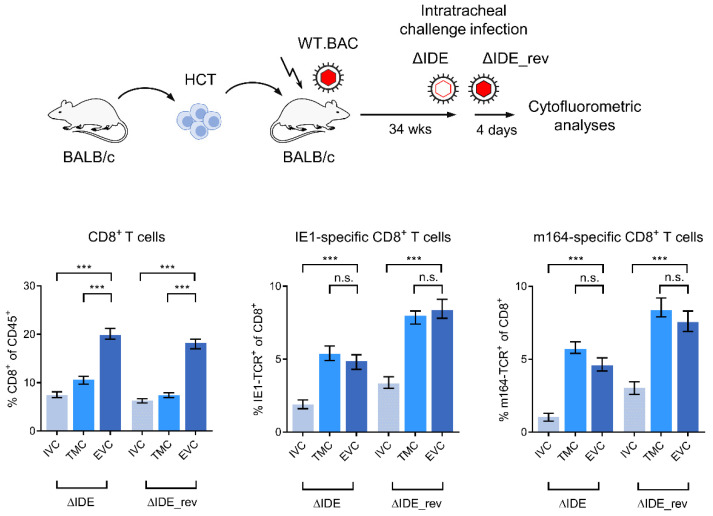
Acute airway superinfection recruits CD8^+^ T cells into TMC and EVC in an epitope-unspecific and an epitope-specific manner. (Top) Scheme of the experimental protocol used to test for viral epitope-specific recruitment by airway challenge infection. (Bottom) Summary of relative cytofluorometric quantitations. Bars represent median values, and error bars represent the range of 4 pools from 4 mice each (*n* = 4) in the case of IVC and TMC, and, owing to low cell yield, of 2 pools from 8 mice each (*n* = 2) in the case of the EVC. See the body of the text for further explanation. Significance levels are shown for groups of most interest. (***) *p* < 0.001. (n.s.) not significant, *p* > 0.05.

## Data Availability

The data presented in this study are available on request from the corresponding author.

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
