# Peer review of "Localization of Viral Epitope-Specific CD8 T Cells during Cytomegalovirus Latency in the Lungs and Recruitment to Lung Parenchyma by Airway Challenge Infection"

_life, 2021, doi:10.3390/life11090918_

Round 1
Reviewer 1 Report
Memory inflation (MI) is a phenomenon that virus-specific CD8+ T effector-memory cells (TEM) are expanded in persisting lung infiltrates during the establishment of cytomegalovirus latent infection. The characteristic of this event is the emergence and expansion of a CD8+ T cell subset defined by the phenotype KLRG1+ CD62L-. In this brief report from Blaum et al, the authors observed that the frequencies of viral epitope-specific inflationary T effector-memory cells (iTEM) were nearly equivalent in the IVC and TMC compartments, but were reduced in the EVC compartment. This reduction was correlated with an increase in KLRG1-CD62L- conventional T effector-memory cells (cTEM) and a decrease in functional IFNγ+ CD8+ T cells. They further showed that this iTEM defect in EVC compartment was restored by providing viral antigen through acute airway superinfection.
In general, this manuscript is well written and the results are clear and well presented. The findings are interesting in the CMV field. However, before the conclusions can be seen as sound and appropriate, the statistical significance analysis has to be performed throughout the data, specifically in Fig. 2, Fig. 4, and Fig. 5.
Author Response
We thank the reviewer for helpful comments and suggestion for improvement. As suggested, significance levels are now included in Figures 4 and 5 in the comparisons of groups of most interest. Data in Figure 2 result from a linear regression analyzes and the 95% confidence intervals are indicated. it is explained in the legend the differences are significant when the confidence intervals are not overlap.
Reviewer 2 Report
This is a very well written manuscript detailing the use of a transwell system to study a very complicated interaction between tissue resident CD8-TCells and the airway epithelial cells. While the results are not exactly as predicted by their original hypothesis, the authors put forward a good discussion to try and explain their findings. These studies need to be done even more to be able to understand the true interaction between tissue resident immune cells and how they protect us from viral infections.
Author Response
Thank you for helpful comments. We agree that future work should be undertaken to further enhance our understanding of the interaction between pulmonary T cells and infected lung cells in the control of infection.